# Therapeutic Efficacy of an Erythromycin-Loaded Coaxial Nanofiber Coating in a Rat Model of *S. aureus*-Induced Periprosthetic Joint Infection

**DOI:** 10.3390/ijms25147926

**Published:** 2024-07-19

**Authors:** David C. Markel, Dexter Powell, Bin Wu, Paula Pawlitz, Therese Bou-Akl, Liang Chen, Tong Shi, Weiping Ren

**Affiliations:** 1The CORE Institute, 26750 Providence Pkwy #200, Novi, MI 48374, USA; david.markel@thecoreinstitute.com; 2Department of Biomedical Engineering, Wayne State University, 818 W. Hancock, Detroit, MI 48201, USA; liangchen89@gmail.com (L.C.); ts1730us@yahoo.com (T.S.); 3Section of Orthopaedic Surgery, Ascension Providence Hospital Orthopaedic Research Laboratory, 16001 West Nine Mile Road, Southfield, MI 48075, USA; dexter.powell@ascension.org (D.P.); binwu777@gmail.com (B.W.); paula.dietz@ascension.org (P.P.); wren1952@gmail.com (W.R.); 4John D. Dingell VA Medical Center, 4646 John R St., Detroit, MI 48201, USA

**Keywords:** coaxial nanofibers, erythromycin (EM), periprosthetic joint infection (PJI), surface coating, drug release, osteointegration, rat model

## Abstract

Implant surface nanofiber (NF) coatings represent an alternative way to prevent/treat periprosthetic joint infection (PJI) via local drug release. We developed and characterized a coaxial erythromycin (EM)-doped PLGA/PCL-PVA NF coating. The purpose of this study was to determine the efficacy of EM-NF coatings (EM0, no EM, EM100 (100 mg/mL), and EM1000 (1000 mg/mL) wt/wt) in a rat PJI model. A strong bond of the EM-NF coating to the surface of titanium (Ti) pins was confirmed by in vitro mechanical testing. Micro-computed tomography (mCT) analysis showed that both EM100 and EM1000 NF effectively reduced periprosthetic osteolysis compared to EM0 at 8 and 16 weeks after implantation. Histology showed that EM100 and EM1000 coatings effectively controlled infection and enhanced periprosthetic new bone formation. The bone implant contact (BIC) of EM100 (35.08%) was higher than negative controls and EM0 (3.43% and 0%, respectively). The bone area fraction occupancy (BAFO) of EM100 (0.63 mm^2^) was greater than controls and EM0 (0.390 mm^2^ and 0.0 mm^2^, respectively). The BAFO of EM100 was higher than that of EM1000 (0.3 mm^2^). These findings may provide a basis for a new implant surface fabrication strategy aimed at reducing the risks of defective osseointegration and PJI.

## 1. Introduction

Septic and aseptic loosening are common failure mechanisms of total joint arthroplasty (TJA) [1]. Improving osseointegration and preventing periprosthetic joint infection (PJI) have great clinical benefits. An orthopedic implant that would promote rapid osseointegration and prevent PJI would be of great benefit, particularly when placed into bone compromised by disease or physiology.

Treatment of PJI is complex and the use of antibiotics such as ceftriaxone, clindamycin, and vancomycin is based on the bacterial strain found in synovial fluid, blood culture, and periprosthetic tissue [2]. The recommendations and guidelines for PJI treatments vary among international societies of infectious diseases, but generally high doses and long-term intravenous and/or oral treatment with surgical debridement are recommended to eradicate the PJI and retain the implant. Sometimes combinations of two or more antibiotics are used as well [2,3]. Despite the effectiveness of these standard treatments, they still carry some risks. Some complications are medical and related to the type of antibiotics used, like acute kidney injury or drug-induced hepatitis [3,4]. Other complications are related to the IV catheter, like bacteremia, deep vein thrombosis (DVT), occlusion, and longer hospital readmission [3]. To minimize the complications and improve patients’ recovery, newer strategies have been tried, such as the local administration of antibiotics using catheters with or without a carrier, like bone cement polymethylmethacrylate (PMMA) beads or a spacer [5,6]. Currently, PMMA is the most used carrier, but one of its downsides is that it is non-degradable and requires complete removal after completing the treatment. There is an increased interest in biodegradable materials capable of sustained drug release for PJI treatment. Numerous strategies have been attempted to prevent implant infection by either implant surface fabrication or incorporation of antibiotics into the implant devices [7]. The use of hydroxyapatite (HA) to stimulate bone growth is popular due to its biocompatibility and resorptive nature [8,9]. Use of HA coatings as a drug delivery agent is limited since there is burst drug release due to weak binding between the loaded drug and the HA surface [10]. Recent developments in material science have shown that implants with biodegradable polymer coatings may be used for controllable and sustained antibiotic delivery [11]. This minimizes local or systemic toxicity associated with high fluctuating antibiotic concentrations [7]. The antibiotic-eluting polymer coatings may have limitations such as chemical instability and/or local inflammatory reaction. These limitations may be potentially compensated for by combining the polymer with a nanostructured coating [11]. Based on an extensive literature search, few investigations have been made in this field [12]. The proposed study is expected to fill this gap. We believe that a high EM loading and a sustained EM delivery from NF coating are reachable via tailoring the formulation of NF microstructures.

Erythromycin (EM), a commonly used antibiotic, is a 14-membered lactone ring macrolide. It is effective against most bacteria encountered in PJI [13,14]. EM has other biological effects well beyond its antimicrobial activity [14]. Current studies, including our own [15], indicated that EM exerts its biological effects through targeting of NF-kB signaling [16]. There are several drug candidates for osseointegration that are under active investigation, including bisphosphonate, bone morphogenetic proteins (BMPs), statins, and antagonists of TNF and IL-1 (e.g., etanercept, infliximab, and anakinra). Unfortunately, the unwanted side effects are of clinical concern. EM is effective in ameliorating chronic inflammation [15] and inhibiting wear debris-induced inflammation and bone loss (both basic science and clinical trial) [16] by inhibiting osteoclast formation and enhancing osteoblast activity. The effects are immunomodulatory rather than immunosuppressive [17]. This is important since PJI patients have some immunological dysfunction [18]. In addition, EM has a positive anabolic effect and has been shown to positively impact osteoblast growth and differentiation of murine MC3T3 pre-osteoblasts [19]. The challenge is delivering EM to periprosthetic tissue in a sustained and predictable way [20].

Implant surface fabrication techniques have been developed to improve osseointegration and prevent PJI via sustained periprosthetic drug delivery [7]. One of the most promising methods of modifying an implant surface is nanoscale coating via electrospinning [21]. By using coaxial electrospinning [22], nanofibers (NFs) can be applied to an implant and used as controllable drug release devices. In a coaxial core–sheath system, drug release is affected by both the concentration gradient and the degradation rate of the sheath barrier [23]. The advantages of electrospun NFs include a high surface area, high mass-to-volume ratio, and a small inter-fibrous pore size with high porosity [24]. The potential application of NFs for the enhancement of osseointegration is promising but often overlooked. There are a few papers [25,26,27] demonstrating that the NF structural cues alone can be used to create an osteogenic environment and that the cell attachment, proliferation, and differentiation of bone cells are influenced by the physiochemical properties of the NFs [26]. Electrospun NFs can be used as drug release devices by direct blending of drugs into the polymer solution before electrospinning, or via the utilization of coaxial electrospinning [22]. During coaxial electrospinning, a spinneret is employed to trap a secondary fluid layer (containing labile drugs) within the core of the forming NFs [22]. The sheath solution acts as a guide and surrounds the core material. The sheath structure represents a physical barrier to reduce the initial burst release and protects the drugs in the core fiber. The concentration gradient inside the core fiber is the driving force for diffusion [23]. Therefore, control of the drug release rate can be achieved by preparation of various formulations and thicknesses of slow/fast-degradation sheath fibers and/or modification of physiochemical properties of NFs [23].

We developed a coaxial electrospun NF composed of polycaprolactone (PCL)/poly(lactide-co-glycolide) (PLGA)^sheath^ and polyvinyl alcohol (PVA)^core^ polymers (PCL/PLGA-PVA) [28]. PCL and PLGA are FDA-approved hydrophobic polymers [29] The degradation rate of the PCL/PLGA sheath fiber can be used to control the release pattern of drugs embedded in the PVA core fiber matrix [28,29]. PVA is a water-soluble and biodegradable polymer with good electrospun NF-forming capability due to its excellent electroconductivity [30]. We previously developed electrospun PVA/Collagen/HA NFs and found that the inclusion of HA and Collagen (Col) in the PVA NFs significantly increased the fiber stability and the mechanical strength. The encapsulated nano-HA and Col also enhanced the adhesion and proliferation of osteoblastic MC3T3 cells in vitro. However, these blended PVA/Col/HA NFs cannot be used as a desired drug release device because of their fast degradation rate (~10 days) [30]. In addition, PLGA and PCL can be strongly bonded to a titanium (Ti) implant surface [31]. It has been shown that EM-doped PCL/PLGA-PVA NFs (EM-NF) can be directly deposited onto a titanium (Ti) implant surface during electrospinning [32]. The core and sheath components of the coaxial PCL/PLGA-PVA NF coatings were loaded with EM at different concentrations for this study: EM0 (no EM), EM100 (100 μg/mL), EM500 (500 μg/mL), and EM1000 (1000 μg/mL). A sustained EM release from EM-NFs for >4 weeks was observed in vitro [32]. The eluents collected from EM-NFs showed a strong zone of inhibition (ZOI) to *S. aureus* growth, and the sizes of ZOI positively related to the amount of EM released. In addition, EM-NFs were nontoxic to rat bone marrow stem cells (rBMSCs) and stimulated cell growth and osteoblastic differentiation of rBMSCs [32]. The purpose of this continuing study was to evaluate the in vivo therapeutic efficacy of the EM-NF coating in a rat-infected tibia implantation PJI model. We hypothesize that sustained release of EM from the NF coating will inhibit implant infection and further promote osseointegration due to its proven osteogenic and bacteriostatic or bactericidal activities.

## 2. Results

### 2.1. Bonding Strength of EM-NF Coating on the Surface of Ti Pins

Using an ex vivo porcine bone implantation model, we found that the EM-NF coating remained intact after the test of push in and pull out (Figure 1A,B). A few samples of EM-NF coating were disrupted at the end of the pin that may have been caused by the uneven or rough surface of the created bone holes. The maximum shear force required for insertion of Ti pins coated with EM0, EM100, and EM1000 was 4.51 N, 14.3 N, and 7.8 N, respectively. The maximum shear force required for pull out of Ti pins coated with EM0, EM100, and EM1000 was 2.2 N, 4.9 N, and 4.0 N, respectively (Figure 1C). EM doping increased the force required for both push in and pull out of Ti pins, especially for EM100 (*p* < 0.05). Our data confirmed that the bonding strength of the EM-NFs to the smooth surface of the Ti pins was acceptable for the planned animal experiment.

### 2.2. Animal Study

No rats were dropped from the study due to body deterioration (weight loss or fever), wound infection (swelling of the leg, loss of active motion in knee and ankle joints), or other complications.

### 2.3. μCT Evaluation of Osteolysis

The area of periprosthetic osteolysis was considered to reflect efficacy of infection control. Our results reflect the area of osteolysis around the pin at 8 and 16 weeks based on µCT analysis. The average for each group is plotted and the error bars denote standard deviation. Some observations were made. As expected, in the negative control group there is far less osteolysis as compared to the positive control, EM0. In group 2, EM0, the area of osteolysis continues to enlarge from 8 to 16 weeks due to the untreated periprosthetic infection, whereas in the negative control and treatment groups there is improvement in the lytic area of bone surrounding the pin over time. In summary, EM-NF coating (EM100 and EM1000) effectively reduced *S. aureus*-induced periprosthetic osteolysis vs. EM0 control at both 8 and 16 weeks (Figure 2). At 8 weeks, areas of osteolysis for EM100 and EM1000 were smaller than for the EM0 control and close to the negative control. The trend was even more significant at 16 weeks. Areas of osteolysis were smaller in EM100 than with EM0 but that was not statistically significant (*p* = 0.08).

### 2.4. Histological Analysis of Decalcified Paraffin Tissue Sections

The inhibition of infection and osteolysis by EM-NFs was confirmed by histology (Figure 3). Characteristic features of periprosthetic infection with osteolysis were seen in EM0: sequestrum, adjacent soft tissue abscesses, expanded bone loss around implants, significant bacterial growth, inflammation, and exudation. Treatment with both EM100 and EM1000 controlled local infection and enhanced periprosthetic new bone formation at 8 and 16 weeks. Some EM-NF matrix residue was observed at the interface of newly formed bone and the Ti pin surface at 8 weeks, but none was found at 16 weeks, indicating complete resorption of the NFs.

### 2.5. Hard-Tissue Section Histology

The efficacy of EM100 and EM1000 coatings was further confirmed by 16-week hard-tissue sections. The bone implant contact (BIC) and bone area fraction occupancy (BAFO) within 200 µm of the implanted Ti pin surfaces were used to evaluate the osseointegration around the implants (Figure 4, Table 1). As expected, there was no bone in contact with the implant and no periprosthetic bone surrounding the pin in the positive control, as there was just a large black purulent abscess seen adjacent to the pin hole (the pin was lost because of loosening due to infection). In the erythromycin–nanofiber groups, there was trabecular bone surrounding the implants, in a pattern like that of the negative control group. This bone formation was much denser in the EM100 group, which is reflected in its higher BIC and BAFO values. The EM1000 group did have periprosthetic bone formation, but none directly in contact with the implant surface, and thus a BIC of 0 mm^2^. The BIC (%) of EM100 (35.08%) was higher than that of the negative control (3.43%) and EM0 (0%). The BAFO of EM100 (0.63 mm^2^) was higher than that of negative control (0.390 mm^2^) and EM0 (0.0 mm^2^). The BAFO of EM100 was also higher than that of EM1000 (0.3 mm^2^).

## 3. Discussion

Strategies aimed at reducing PJI and the resultant lack of osseointegration should improve the success of TJA and increase implant longevity [33]. It has been shown in vitro that EM-eluting coaxial PCL/PLGA-PVA NF coating (EM-NF) had a sustained and controlled drug delivery with prolonged bacterial growth inhibition, and the degradation of the NFs matches the period of an implant’s osseointegration [32]. Our data confirmed that the bonding strength of the EM-NFs to the smooth surface of the Ti pins was strong using an ex vivo porcine bone implantation model (Figure 1). The exact mechanism(s) of interfacial adhesion between the NF coating and Ti implant surface is not very clear. One potential reason for this is that PLGA has a much higher ratio of oxygen atoms in its molecular structure than PCL, thus providing more electrostatic interaction on the Ti surface [34]. In addition, the high surface area and high volume-to-mass ratio of NFs might also contribute to the bonding of NFs to the Ti implant surface [35]. We also reported that EM doping increased the bonding strength of EM-NF with Ti pins, especially for EM100 (*p* < 0.05). We believe that might be due to the changes in NF microstructure, wettability, and mechanical properties caused by EM incorporation, as we reported before [32]. In this study, it was demonstrated that the EM-NF coating could be strongly bound to a titanium surface and that the drug elution inhibited periprosthetic infection and enhanced osseointegration up to 16 weeks after implantation in a rat PJI model. The data indicated that the structural and mechanical advantages of the EM-NF coating show the potential to both prevent PJI and enhance osseointegration.

Implant coatings like hydroxyapatite (HA) have been widely used to promote increased implant osseointegration. However, drug delivery from HA is limited due to weak binding between the loaded drug and the HA surface, and a resultant burst drug release [36,37]. PMMA cement has also been applied for antibiotic drug delivery and widely used for PJI prevention and treatment. However, the efficacy of antibiotic-impregnated cement continues to be debated [37,38].

A drug-eluting implant coating should deliver antibiotics at a level well above the minimum inhibitory concentration (MIC) for at least 6 weeks for PJI treatment [10]. The anabolic effect of released EM from NFs is expected to be evident within 4–8 weeks, equivalent to the end of the bone healing formative phase. In this study, we extended our observation by including a bone modeling and remodeling phase (up to 16 weeks). It is crucial to determine whether EM-loaded NF coatings are still effective in inhibiting infection up to 16 weeks during the bone remodeling phase. In our previous study, we noted a constant and sustained EM release from NFs up to 12 weeks when EM was embedded both in the core and sheath fiber solution before electrospinning [32]. The amount of EM released was proportional to the amount of EM doped [32]. The concentration of EM (EM100 and EM1000) released was higher than the concentration required for the MIC of *S. aureus* (1 μg/mL) [39] at all phases [40]. Furthermore, the EM release was stable and inhibited bacterial growth in vitro [32]. Therefore, EM-NF coating is expected to provide a first-line defense against infection particularly if an implant were placed in a contaminated or high-risk surgical field.

We then evaluated the therapeutic efficacy of EM-NF coating in a rat PJI model [41]. This rat model has been widely used for the therapeutic efficacy evaluation of periprosthetic drug delivery [42] and implant surface modification [43]. The rat model has an orthotropic weight-bearing implant being subjected to sustained mechanical shear stresses in vivo. As shown in Figure 2, EM-NF coating (EM100 and EM1000) effectively reduced *S. aureus* infection-induced periprosthetic osteolysis compared to EM0 control at 8 and 16 weeks by μCT analysis. The rate, quantity, and quality of periprosthetic new bone formation (osseointegration) are relied on for efficacy of infection control, as well as the physiochemical and biological properties of EM-NF coating [25,27]. Consistent with the µCT findings, histological analysis of paraffin tissue sections (Figure 3) showed typical features of periprosthetic infection with osteolysis in the EM0 group. Treatment with both EM100 and EM1000 effectively controlled local infection and enhanced periprosthetic new bone formation at both 8 and 16 weeks. The EM-NF were biocompatible and completely resorbed over the study period. The therapeutic efficacies of EM-NF coating were further validated by the measurement of periprosthetic bone volume at 16 weeks (Figure 4, Table 2).

Taken together, data from the animal study demonstrated that both EM100 and EM1000 effectively inhibited *S. aureus* infection and enhanced osseointegration up to 16 weeks. However, EM1000 had lesser impacts on periprosthetic new bone formation compared to EM100, due to the local much higher EM concentration than that of EM100. The current finding agrees with our previous in vitro data that showed EM100 NF coating significantly enhanced the cell growth and differentiation of rat bone marrow cells, which was significantly diminished by EM1000 treatment [32].

There were some study limitations. First, additional microbiological analysis of implanted Ti pins and/or bone marrow cavity washout should be studied in the future to establish the status of infection control (bacterial growth and biofilm formation, etc.). Second, we need to develop noninvasive imaging techniques for the real-time evaluation of in vivo EM release kinetics. Additionally, performing special Gram staining of the histologic sections for *S. aureus* could be helpful to confirm that the infection is caused by this strain. Another limitation in the study was the use of female rats, which could affect bone modeling due to hormonal changes. Finally, the sample size was small (n = 8) when considering the requirements of quantitative histological analysis for both hard-tissue and paraffin section specimens.

## 4. Materials and Methods

### 4.1. Materials

PVA (MW~205,000), PCL (MW~70,000–90,000), PLGA (MW~54,000–69,000), and EM were purchased from Sigma-Aldrich (St. Louis, MO, USA). Titanium pins (0.8 mm diameter, 0.8 mm length with flat heads 1.94 mm diameter) were obtained from Shanghai Sixth Hospital (Shanghai, China). Sprague Dawley (SD) rats (female, BW 200 g–300 g) were purchased from Charles River laboratories (Wilmington, MA, USA). *Staphylococcus aureus* (SA) was obtained from ATCC (#49230). Mueller–Hinton broth (Cat. No. CM0405, Oxoid, Thermo Fisher Scientific, MA, USA).

### 4.2. Preparation of EM- Doped Coaxial PCL/PLGA (1:1)-PVA NFs

The PCL/PLGA solution was prepared by mixing 11% PCL and 15% PLGA (dissolved in chloroform/dimethylformamide (DMF) at the ratio 1:1 (*v*/*v*) with stirring) [32]. PVA 15% solution was prepared by dissolving PVA powders in distilled water at 90 °C. Both PLGA/PCL (sheath) and PVA (core) solution were mixed with EM stock solution (5 mg/mL) before electrospinning. The final EM concentration in the sheath and core fibers were either 100 μg/mL (EM100) or 1000 μg/mL (EM1000). NFs without EM doping (EM0) were included as vehicle controls. The selection of the two EM concentrations (EM100 or EM1000) was based on the in vitro physiochemical and biological characterization of EM-NF formulations [32].

Coaxial electrospinning was performed using a custom-made coaxial nozzle consisting of a hollow stainless-steel T-junction with a fully penetrating 19-gauge core needle. The PCL/PLGA solution (for the core fiber) and PVA solution (for the sheath) were uploaded to the syringes. The syringes were attached to syringe pumps set at a flow rate (Q) of 1 mL h^−1^ for PCL/PLGA and 0.47 mL h^−1^ for PVA. A 20 kV voltage was applied to the needle from the power supply. The distance from the needle tip to the collector was 10 cm [40].

To achieve EM-NF coating, Ti pins were fixed on a specifically designed collector, which has two horizontal and parallel embedded and electric-conductive sticks to hold the Ti pins. EM-NFs were directly deposited on the Ti surface. The coated pins were then soaked briefly in 70% ethanol, air-dried, and sterilized by UV light irradiation overnight before animal experiments. NF-embedded EM is still active within the coating after treatment as manifested by microbiological assays (bacterial culture inhibition assay). The thickness of the EM-NF coating (~50 μm) was determined by a micro-caliper.

### 4.3. Bonding Strength of Ti Pins with EM-NF Coating

An ex vivo porcine bone implantation model [33] was used to determine whether the EM-NF coating remained intact during implantation. Fresh-frozen porcine knee specimens were purchased from a local slaughterhouse and stored at −20 °C. The specimens were thawed overnight, and all soft tissues were removed before testing. A flat dissected bony surface was obtained after removing all superficial articular tissues. Tibial bone holes (diameter = 0.835 mm; depth = 10 mm) were made. Ti pins with EM-NF coating were slowly inserted into the bony holes using an Instron model 8841 Universal Materials Test Machine at the speed of 3 mm min^−1^. The porcine bones with implanted Ti pins were then soaked in sterilized PBS at 37 °C for two days before the pull out of the implanted Ti pins. The maximum friction force required for both push in and pull out of Ti pins were recorded (n = 3).

### 4.4. Rat Model of S. aureus-Infected Tibia Implantation

*S. aureus* bacterial suspension preparation: A bacterial suspension (inoculum) was created by growing the bacteria overnight in Mueller–Hinton broth in a shaking water bath at 37 °C. The resultant bacterial solution concentration was determined by placing a 1.5 uL sample onto a NanoDrop spectrophotometer (Thermo Scientific) and measuring the optical density (OD) at 600 nm in triplicate. This was compared to a standard curve with known concentrations.

Animal surgery: The study was approved by Ascension Providence Hospital, Institutional Animal Care and Use Committee (IACU) protocol # 101-19, and the authors adhered to the ARRIVE guidelines and have supplied the Checklist. Fifty-six (56) female Sprague Dawley (SD) rats were divided into four groups (n = 8 per group and each time point, Table 2) and sample size was chosen based on previous experience and similar studies. Rats were anesthetized by intraperitoneal injection (Xylazine at 8 mg/kg and Ketamine at 75 mg/kg). The right hind limb area was shaved and prepped with betadine and 70% alcohol. A 10 mm medial parapatellar incision was made and the patellar tendon laterally dislocated to expose the tibial plateau. The center of the tibial plateau was reamed to form a circular indent of 1.5 mm diameter and 0.5 mm depth. A pilot hole was created at the intercondylar eminence and a 1.0 mm Ti wire used to make a channel into the medullary canal. Pre-colonized NF coated Ti pins were inserted into the medullary canal to form a part of the knee joint. A mixed contamination model of *S. aureus* was used: 10 μL (1 × 10^3^ CFU/mL) of freshly prepared inoculum in broth was directly injected into the medullary cavity before pin insertion followed by one-step pin dip-soak for a few seconds in 2 mL of inoculum with the same concentration (1 × 10^3^ CFU/mL) just before implantation. Wounds were closed in layers and the rats observed until the effects from the anesthetic wore off. Animals were housed individually and provided food and water ad libitum. Buprenorphine 0.01–0.05 mg/kg was injected subcutaneously preoperatively and after 12 h for the first 24 h. Ketoprofen 5 mg/kg was administered postoperatively and once daily for one week. Postoperative care: Anorexia was managed with supplemental food or treats if weight loss was not above or equal to 20% of the original body weight. Hemorrhage did not occur, but animals were monitored before closing to maintain homeostasis. Minor lameness was treated with analgesics. Rats were sacrificed at 8 and 16 weeks after surgery, respectively.

### 4.5. Evaluation of Periprosthetic Osteolysis by Micro-Computed Tomography (μCT)

Baseline scanning was performed immediately postoperatively then at sacrifice (8 or 16 weeks) using a μCT (vivaCT 40, Scanco Medical AG, Brüttisellen, Switzerland) set at 70 kV, 114 μA, with 650 ms integration time and 40 μm resolution. Periprosthetic osteolysis area (mm^2^) was measured on defined sections 0.2 mm from the pin surface (Figure 5A). Briefly, all 2D μCT slices were examined from the top of the pin to the distal end. The image slice with the largest area of periprosthetic osteolysis was selected and the contour around the bony margin of that area was measured with built-in software (Figure 5B). The calculated Ti pin area was 0.0064 cm^2^.

### 4.6. Histological Analysis of Decalcified Paraffin Specimens

Tibia samples were fixed (10% formalin) for 24 h then decalcified (10% ethylene diamine tetra-acetic acid (EDTA)). The Ti pins were gently removed with minimal loss of tissues at the bone–implant interface. Paraffin embedded tissue sections (6 μm) were cut and stained with hematoxylin & eosin (H&E). Bone area (μm^2^) was the measured area of bone tissue in the region 200 µm from the pin surface. Presence of fibrous tissue, infection, and inflammatory cellular infiltration were described. The histology slides were evaluated by an independent investigator blinded to treatment.

### 4.7. Histomorphometry of Hard-Tissue Sections

One tibia containing a pin from each group was fixed in 70% alcohol, dehydrated in graded ethanol solutions, and embedded in methyl methacrylate without decalcification. Sections, 40 μm thick, were created perpendicular to the implant’s long axis using a rotary diamond saw (SP1600, Leica, Germany) then polished (SP2600, Leica, Wetzlar, Germany). The sections were stained with 1% toluidine blue. Bone-to-implant contact (BIC) was calculated as the linear percentage of the interface with direct bone-to-implant contact to total interface of the implant in the cancellous bone. Bone area fraction occupancy (BAFO) was measured as the area percentage of bone tissue to the whole area, defined as a ring 200 μm from the implant surface.

### 4.8. Statistical Analysis

Values are expressed as mean ± standard deviation. All analyses were performed using Microsoft Excel 2016. ANOVA single factor was used for comparison between group means and a *p*-value < 0.05 was considered significant. ANOVA was followed by a Tukey Kramer post hoc test if necessary. When descriptive statistics determined data were not normally distributed, analysis between groups was performed using Kruskal Wallis followed by Dunn’s post hoc test if necessary.

## 5. Conclusions

Data from this animal study demonstrated that EM-NF coatings could be well bound to a titanium surface and were dually functional in enhancing osseointegration and eliminating bone infection up to 16 weeks. These findings may provide a new implant surface fabrication strategy aimed at reducing the risks of defective osseointegration and PJI, and likely to improve the success rate of TJR.

## Figures and Tables

**Figure 1 ijms-25-07926-f001:**
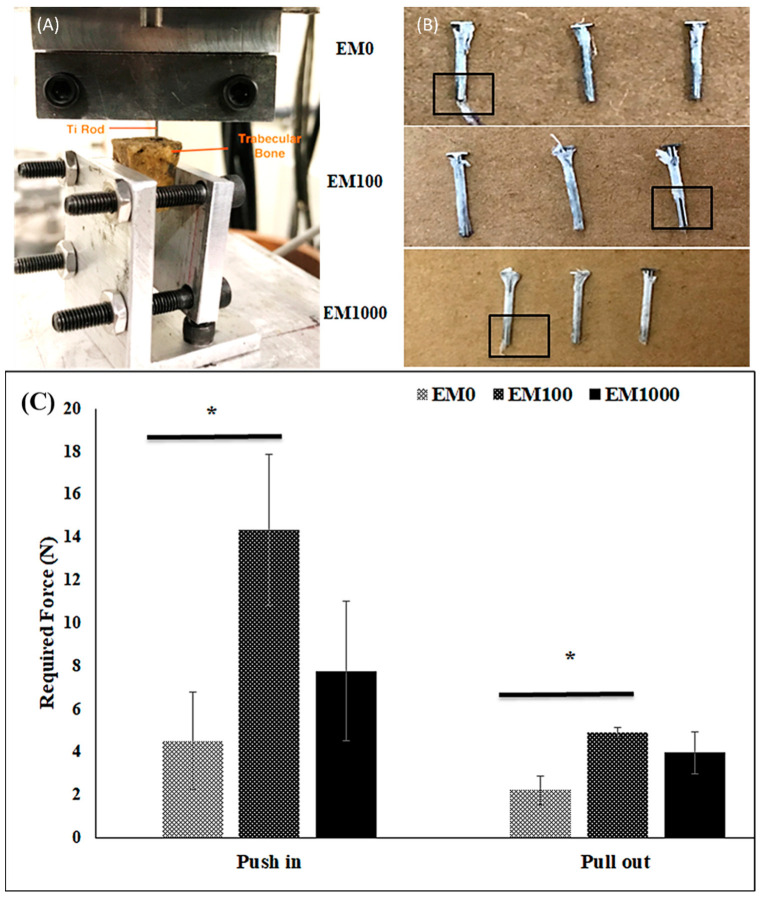
Ex vivo mechanical testing using a porcine bone implantation model: (**A**) the mechanical testing set up; (**B**) Ti pins with EM-NF coatings after push in and pull out test. The broken areas of EM-NF coating after testing were marked by a black box (n = 3). (**C**) Comparison of push in and pull out forces of Ti pins with EM-NF coating (n = 3). * *p* < 0.05 between EM0 and EM100.

**Figure 2 ijms-25-07926-f002:**
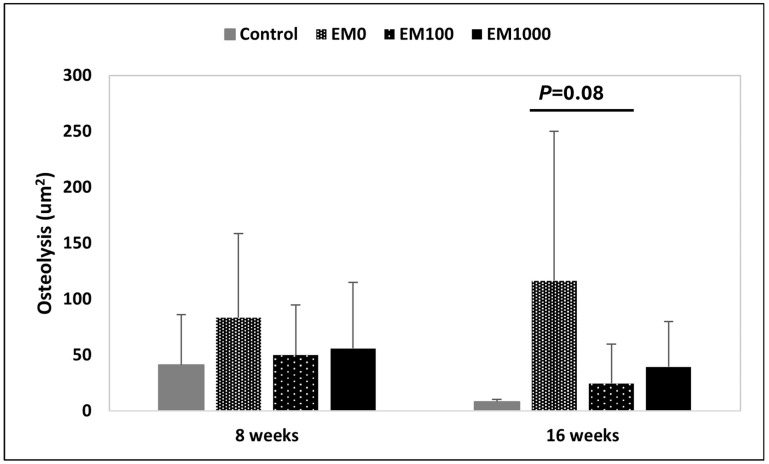
Measurement of periprosthetic osteolysis area (μm^2^). Comparison of osteolysis areas among groups at 8 weeks and 16 weeks. *p* = 0.08 between EM0 and EM100 groups at 16 weeks (n = 8 for each group and each time point).

**Figure 3 ijms-25-07926-f003:**
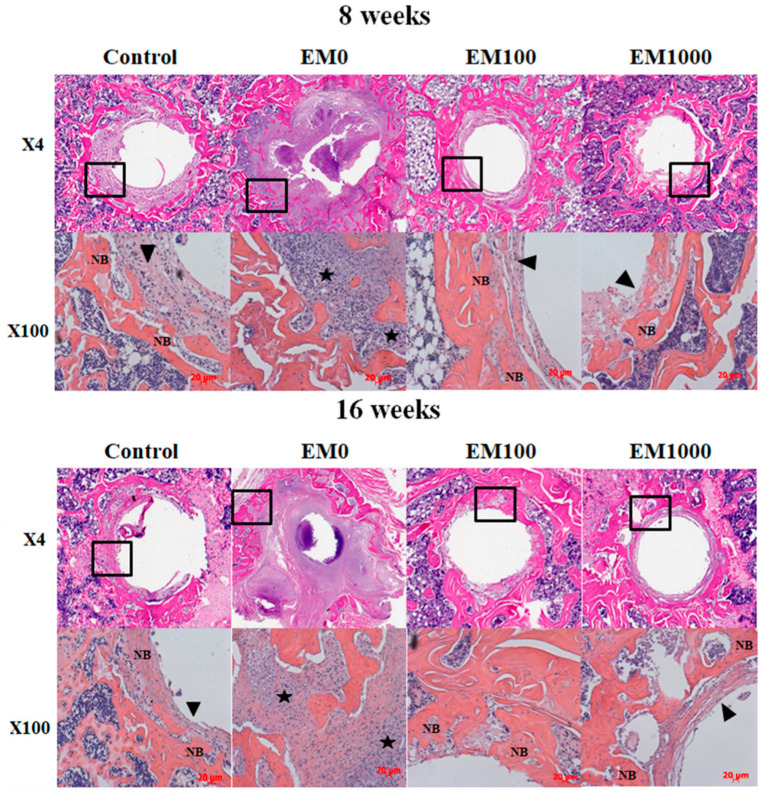
H&E staining of paraffin sections of tibia samples 8 and 16 weeks, respectively, for EM-NFs groups after implantation (n = 5), in which bone area is referred to as the mature bone surrounding the Ti implant. NB, newly formed bones surrounding the Ti implants within 200 µm; black arrowheads, fibrous tissue; black stars, infection and inflammation area.

**Figure 4 ijms-25-07926-f004:**
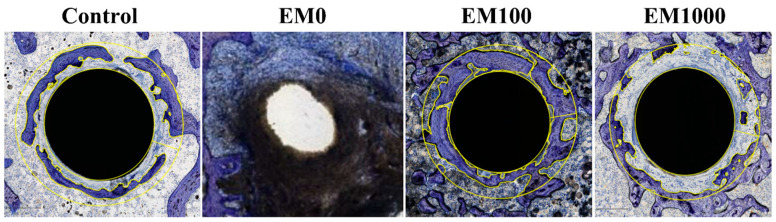
Toluidine blue staining of hard-tissue sectioning collected at 16 weeks after implantation for negative control, EM0, EM100, and EM1000 groups. Bone implant contact (BIC) and bone area fraction occupancy (BAFO) scores within 200 μm of the medullary implant (implant = yellow inner circle; 200 µm ROI from implant = yellow outer circle).

**Figure 5 ijms-25-07926-f005:**
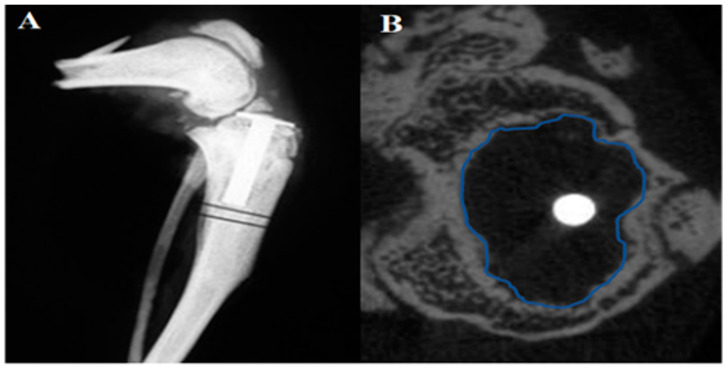
Measurement of periprosthetic osteolysis area. (**A**) µCT scan showing the position of Ti pin implantation. (**B**) Scanning through all slices from the segment under the pin head to the end of the pin was performed and a slice with the largest area of osteolysis was selected. A contour around the cortical bone is outlined, and the area of the contour (include units) is recorded. Pin area is constant and approximately 0.0064 cm^2^.

**Table 1 ijms-25-07926-t001:** BIC and BAFO measurement.

Group	BIC (%)	BAFO	Bone-to-implant contact (BIC): The BIC was calculated as the percentage of implant surface in direct contact with the bone over the entire length of the implant within the section examined. Bone area fraction occupancy (BAFO): The BAFO was the area occupied by the mineralized bone matrix within 200 um of the implant within the section. Measurement was made by outlining the bone surface area from the total field within a 200 um radius of the implant and was expressed as percentage (implant area was removed).
Negative control	3.43	0.39	
EM0	0	0	
EM100	35.08	0.63	
EM1000	0	0.3	

**Table 2 ijms-25-07926-t002:** Rat groups.

Description	8 Weeks	16 Weeks
NF coating, no infection, negative control	4	4
EM0-NF coating, positive control	8	8
EM100-NF coating	8	8
EM1000-NF coating	8	8

## Data Availability

The raw data supporting the conclusions of this article will be made available by the authors on request.

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
