# Peer review of "Therapeutic Efficacy of an Erythromycin-Loaded Coaxial Nanofiber Coating in a Rat Model of S. aureus-Induced Periprosthetic Joint Infection"

_ijms, 2024, doi:10.3390/ijms25147926_

Round 1

Reviewer 1 Report

Comments and Suggestions for Authors

The authors aimed to demonstrate whether erythromycin added to the nanofibers had any activity against prosthetic infections in experimental animals. The manuscript is interesting and well conducted, however there are a couple of things I would like the authors to respond to.

The first thing is the absence of demonstration in the histologic samples of S.aureus. Has a culture been performed to confirm that the infection is caused by this strain?

Certainly the number of animals is small, drawing firm conclusions can be problematic.

The other aspect, although not statistically significant, is that it seems that the more concentrated EM 1000 is more active than the EM100. Is there a probable explanation? Figure 3 should be improved and made more readable.

Minor:

S.aureus should be written in italics

The title should be changed. Since erythromycin is active on gram positives, it should be added in the title that the periprosthetic infection is due to S.aureus.

Reviewer 2 Report

Comments and Suggestions for Authors

Dear Authors, I had the opportunity to review your article, which I largely appreciated, as the topics were presented effectively and scientifically valid. The methodology is solid, as far as I could determine, although some limitations are evident and certainly deserve further investigation, as I will detail below. In particular, the introductory section provides a general overview of the importance of periprosthetic infections and the use of antibiotics. However, a detailed discussion on existing approaches and their specific limitations, such as systemic antibiotics and traditional coatings, is lacking. The choice of erythromycin as an antibiotic and coaxial nanofibers as a delivery method should be better justified with reference to current literature. Therefore, I suggest expanding the literature review to include recent studies on antibiotic delivery methods in orthopedic contexts and to provide a clearer and more detailed rationale for the choice of erythromycin and coaxial nanofibers. The description of the preparation of PCL/PLGA-PVA coaxial nanofibers is incomplete. Details such as polymer concentrations, flow rate, applied voltage, and environmental conditions during electrospinning are crucial for the replicability of the study. Additionally, the preparation of the Staphylococcus aureus solution and its application to the pins require a more precise description. Post-operative pain management in rats is mentioned only briefly. It is essential to provide details on dosages, frequency, and any other animal welfare measures. Therefore, it is suggested to include a comprehensive and detailed description of the electrospinning process, specify exactly how the bacterial solution is prepared and applied, and add a detailed section on the post-operative pain management protocol and animal welfare monitoring. The results are presented in general terms without providing detailed quantitative data. For example, micro-CT and histological results should include mean values, standard deviation, and the number of replicates for each group. The description of the statistical analysis is inadequate. Exact p-values for all comparisons made are not reported. In conclusion, the article presents an interesting study on the use of erythromycin-loaded coaxial nanofibers to reduce periprosthetic osteolysis. However, there are several areas that require significant improvements, as specified in detail above, including a detailed description of the methods, comprehensive presentation of the results, and an in-depth discussion of the study's implications and limitations. A thorough review and greater attention to detail will greatly enhance the quality and impact of the work.

Round 2

Reviewer 2 Report

Comments and Suggestions for Authors

Dear Authors, I have read your additions, finding them entirely consistent and in accordance with my expectations. I consider your work interesting. The purpose is clear and respected. I believe that the information provided is to be considered entirely sufficient and represents useful elements to encourage the development of new scientific work.